# The Influence of Organizational Aspects of the U.S. Agricultural Industry and Socioeconomic and Political Conditions on Farmworkers’ COVID-19 Workplace Safety

**DOI:** 10.3390/ijerph20237138

**Published:** 2023-12-03

**Authors:** Fabiola M. Perez-Lua, Alec M. Chan-Golston, Nancy J. Burke, Maria-Elena De Trinidad Young

**Affiliations:** Department of Public Health, School of Social Sciences, Humanities and Arts, University of California, Merced, 5200 N Lake Road, Merced, CA 95343, USA; achan-golston@ucmerced.edu (A.M.C.-G.); nburke2@ucmerced.edu (N.J.B.);

**Keywords:** Latinx farmworkers, occupational health, occupational safety, political economy, COVID-19

## Abstract

Farmworkers in the U.S. experienced high rates of COVID-19 infection and mortality during the COVID-19 pandemic. Their workplace may have been a significant place of exposure to the novel coronavirus. Using political economy of health theory, this study sought to understand how organizational aspects of the agricultural industry and broader socioeconomic and political conditions shaped farmworkers’ COVID-19 workplace safety during the pandemic. Between July 2020 and April 2021, we conducted and analyzed fourteen in-depth, semi-structured phone interviews with Latinx farmworkers in California. Findings show that regulatory oversight reinforced COVID-19 workplace safety. In the absence of regulatory oversight, the organization of the agricultural industry produced COVID-19 workplace risks for farmworkers; it normalized unsafe working conditions and the worker—rather than employer—responsibility for workplace safety. Under these conditions, farmworkers enacted personal COVID-19 preventative practices but were limited by financial hardships that were exacerbated by the precarious nature of agricultural employment and legal status exclusions from pandemic-related aid. Unsafe workplace conditions negatively impacted workplace camaraderie. Study findings have implications for farmworkers’ individual and collective agency to achieve safe working conditions. Occupational safety interventions must address the organizational aspects that produce workplace health and safety inequities and disempower farmworkers in the workplace.

## 1. Introduction

Epidemiological studies showed that farmworkers in the U.S. were at an increased risk of COVID-19 infection during the COVID-19 pandemic [1,2,3]. COVID-19 infection rates were higher among farmworkers compared to other occupational groups, such as those in the public and retail sectors [1]. Large COVID-19 outbreaks occurred in meat and poultry processing facilities [4,5] and there were high rates of COVID-19 infections among field crop workers [6], suggesting that the agricultural workplace may have been a significant site of exposure to the novel coronavirus [3,7]. Not all agricultural employers enforced the use of masks or social distancing in the workplace [8]. Fast-paced working environments may have undermined COVID-19 workplace safety policies [8]. Employer retaliation and worker mistreatment may have discouraged farmworkers from reporting COVID-19 safety concerns or violations [9]. Little is known, however, about how the organization of the agricultural industry and the broader social, economic, and political conditions may have produced greater workplace risk of COVID-19 infection for agricultural workers and influenced their individual agency to protect themselves from the novel coronavirus at work. We analyzed narrative descriptions of the workplace safety experiences of farmworkers that were drawn from in-depth interviews with farmworkers in California to understand how organizational aspects of the agricultural industry and broader socioeconomic conditions shaped farmworkers’ workplace COVID-19 safety. Whereas epidemiological methods may empower researchers to define and administer population health [10], qualitative methods center the knowledge of the individuals who experience health inequities and result in detailed descriptions of what those experiences entail, including their physical and emotional impacts [11]. Thus, qualitative methods provide a critical method to understand the nuances of health risks among populations that have been identified as high-risk or vulnerable by epidemiological research. In this study, the narratives of farmworkers provide insights into social processes and organizational dynamics and how they shape the phenomenon of disproportionate health impacts of the COVID-19 pandemic on farmworkers [11]. Political economy of health theory directs focus to the underlying structural causes and mechanisms of health disparities; it helps connect the structural elements identified in farmworker narratives to identify systems of oppression that produce health inequities [12]. In the next sections, we introduce the guiding theoretical framework and methodology. We present the results of the study and end with a discussion of the implications of the findings. Respondents in this study self-identified as *Latino* but we use the term *Latinx* because it is widely used in the academic literature [13].

### 1.1. The Political Economy of Farmworker Health

Political economy of health theory is concerned with how political and economic structures produce social inequities [12]. From a political economy of health perspective, organizational infrastructures and policies reflect the broader political and economic context in which health or illness unfold [14]. In capitalist societies, organizational policies, practices, and arrangements that prioritize capital over human health may produce health risks [15]. The intersection of capitalism and racism may compound health risks for people with marginalized ethnic/racial identities [12]. 

By considering historical context, political economy of health theory recognizes that contemporary health inequities are often rooted in past political and economic systems and legacies of racism. In this section, we situate farmworker health in a historical context to reveal the political and economic interests that are embedded in the organization of the agricultural industry and how they have shaped organizational policies, practices, and arrangements that ultimately influence the health of Latinx immigrant farmworkers [12].

The agricultural industry has its roots in plantation slavery of the seventeenth and eighteenth centuries [16]. Growers have continued to benefit from racist policies and legal systems long after the abolition of slavery. Eighteenth and nineteenth century Jim Crow policies such as sharecropping, peonage, tenant cropping, and vagrancy laws disproportionately sentenced Black and Latinx individuals to work off fines or debts in the form of agricultural indentured servitude [17,18]. During the New Deal era, southern Democrats in Congress achieved agricultural exemptions from the Fair Standards Act (FLSA) of 1938, which protects against violations such as wage theft, and the National Relations Labor Act (NRLA) of 1935, which established protections for forming unions, to limit protections for Black farmworkers [19]. Limited protections made Black workers vulnerable to coercion, mistreatment, and exploitation, thereby preserving growers’ access to cheap labor [15,19].

U.S. growers have benefited from immigration policies that make immigrant workers vulnerable to coercion and abuse [20,21]. Historically and currently, growers have hired immigrants because these workers have been willing to fill low-paying agricultural jobs, often due to the severe economic conditions in their countries of origin or their exclusion from the U.S. labor market [20]. Throughout the 19th and 20th centuries, growers systematically imported foreign workers from the Philippines, Jamaica, and other countries as laborers with the support of U.S. immigration policy [22]. By the mid-1900s, the U.S. government had created the Bracero Program, which became the largest guestworker program in U.S. history by importing hundreds of thousands of Mexican workers across the U.S.–Mexico border to labor on U.S. farms [23]. Bracero workers, with their precarious legal status, were vulnerable to mistreatment and abuse from growers and neglected by regulatory health and safety policies [20,22,23].

Today, the U.S. agricultural industry continues to rely on the labor of Latinx immigrant workers. Approximately 70% of the hired crop farm labor force in the U.S. is foreign-born [24]. Over half of all hired farmworkers in the U.S. are undocumented Mexican immigrants and a growing number are H2-A visa workers [24]. Undocumented immigrant farmworkers lack legal authorization to live and work in the U.S. and are excluded from the labor market [25]. The H2-A visa program, which is the most recent iteration of the Bracero program [22], allows growers to hire foreign workers to fill temporary jobs in agriculture [26]. H2-A visa workers’ legal status and authorization are legally bound to their employment [26]. Being an undocumented or H2-A visa worker enhances vulnerability to worker mistreatment and severe exploitation [27,28]. Undocumented farmworkers have reported wage theft concerns [29,30] and H2-A farmworkers have reported abuse and mistreatment by employers, including debt bondage [31]. Both undocumented and H2-A farmworkers may be hesitant to report labor and workplace safety violations due to fear of deportation and job loss [32,33]. 

Indeed, a brief review of the history of the U.S. agricultural industry illustrates how the economic interests of growers and racist government policies forged the contemporary organization of the agricultural industry and workplace [15]. Thus, it is critical for the advancement of farmworker health to understand the workplace COVID-19 safety experiences of Latinx immigrant farmworkers in the context of the organization of the agricultural industry and broader socioeconomic and political conditions that make Latinx immigrant farmworkers vulnerable to industry exploitation. Such an understanding will elucidate how the industry may have produced risk of workplace COVID-19 exposure for farmworkers and influenced their individual agency to protect themselves. 

### 1.2. Latinx Farmworkers and COVID-19

The COVID-19 pandemic exposed racial and social inequities in occupational health and safety that negatively impacted Latinx immigrant farmworkers despite their essential worker status [34,35,36,37]. Political economy of health theory suggests that the organization of the agricultural industry likely produced workplace safety inequities during the COVID-19 pandemic that shaped Latinx immigrant farmworkers’ workplace risk of exposure to the novel coronavirus. Yet, the research on the occupational health and safety of Latinx farmworkers during the pandemic has largely focused on assessing morbidity and mortality trends in COVID-19 [1,4,7,38,39]; identifying individual-level risk factors of COVID-19-related physical and mental health outcomes [2,6,9,40]; determining barriers to COVID-19 testing and vaccination [41,42]; and introducing strategies to mitigate the spread of COVID-19 among farmworkers that do not address the structural conditions that may produce risk of exposure [43,44]. Individualized approaches to understanding farmworker safety presume individual responsibility for occupational health and safety and may inadvertently place blame on individuals. Yet, farmworkers have always resisted unsafe working conditions, from social movements (e.g., strikes, boycotts, and marches) [45,46] to individual acts of defiance (e.g., refusing to work in extreme heat) [47,48]. Political economy of health theory can go beyond the individualized focus on personal COVID-19 risk factors among farmworkers to address the organizational factors, power dynamics, and economic forces that likely limit their capacity to exercise agency in ways that improve their workplace safety [12]. While some studies have identified agricultural employers and supervisors as significant shapers of workplace safety during the pandemic [8,49], there has been a missed opportunity to contextualize employer and supervisor practices within the organization of the agricultural industry to examine the role of the industry in shaping occupational health and safety inequities. This study seeks to address these gaps in the literature by applying the political economy of health theory to the study of farmworkers’ workplace safety during the COVID-19 pandemic. We present our data and methods in the next section.

## 2. Materials and Methods

### 2.1. Positionality Statement

The study was informed by the first author's experiences with immigration policy and the agricultural industry; the co-authors' expertise in qualitative methodologies and immigration policy; and the research team’s intimate knowledge of the study region. Perez-Lua is a first-generation, Latinx, and bilingual researcher who was born and raised in the study region and studies the health impacts of immigration policy and the agricultural industry. Chan-Golston is a researcher with a mixed ethnic/racial identity (White and East Asian) who was born and raised in the study region and is an expert in quantitative methodology. Burke is a White, bilingual medical anthropologist whose work focuses on the health impacts of structural racism and urbanism. Young is a White, Latinx, and bilingual researcher who lives in the study region and studies immigration policy and immigrant health. The study team is committed to equity and justice for farmworkers and immigrants. Honoring the voices of Latinxs during the pandemic was at the core of the study.

### 2.2. Data

Data for this study was drawn from in-depth semi-structured phone interviews (*n* = 39) that were conducted as part of the COVID-19 and Latinx Immigrants in Rural California (CLIMA) study [50]. The CLIMA study sought to understand the social and economic impacts of the COVID-19 pandemic on Latinx immigrants living in rural California [50]. Interviews were conducted with Latinx immigrants between July 2020 and April 2021. 

For the current study, we analyzed a subset of 14 interviews from respondents that were employed in the agricultural industry at the time of the study. We limited the sample to respondents who were employed in the agricultural industry at the time of the study to focus on experiences of farmworkers specifically during the COVID-19 pandemic. This study was approved by the University of California, Merced’s Institutional Review Board.

### 2.3. The CLIMA Study

The CLIMA study used a community-engaged approach [51] to guide the study design and participant recruitment process. The study team invited Latinx serving organizations in California to join the study’s community advisory board, including farmworker serving organizations such as Farmworker Justice and Pesticide Reform. The research team and advisory board determined the county sample, developed the interview guide, and recruited study participants. Farmworker health was a major area of focus among the organizations serving on the CLIMA community advisory board. 

The research team and advisory board identified four rural counties in California with large populations of Latinx immigrants from which to sample: Merced, Fresno, Imperial, and Tulare. Merced, Fresno, and Tulare counties are located in the heart of California’s San Joaquin Valley, the stretch of agricultural land that runs 400 miles through the middle of the state. The counties are top producers of agricultural commodities [52]. The agricultural labor force in all four counties is predominantly Latinx and immigrant. Demographic data confirmed these four counties had non-metropolitan areas that met the U.S. Census’s definition of rural (50,000 residents or less) and large Latinx immigrant populations. Latinx individuals who were (1) 18 years or older, (2) immigrants or had at least one foreign-born parent, and (3) lived in a rural town in the study region were eligible to participate in the study. Individuals were recruited through referrals from the advisory board and the research team’s personal networks. 

Potential participants were screened for eligibility by asking participants if they (1) were 18 years or older, (2) lived in a rural town in the study region, (3) identified as Latinx, and (4) were an immigrant or had at least one parent who was an immigrant. Eligible individuals were scheduled for a phone interview with a research team member. Phone interviews were conducted due to pandemic restrictions. After obtaining consent, a research team member used a semi-structured interview guide (Appendix A) to facilitate the interview in English or Spanish depending on the respondent’s preference. Questions focused on participants’ experiences during the pandemic. Specific questions addressed their employment and workplace conditions, job duties, occupational health concerns, and COVID-19 in the workplace. Probes were used to obtain more details about specific topics. A short demographic survey was used to collect sociodemographic data; food security was measured using the USDA’s six-item food security module [53]. Respondents received a $25 e-gift card. Memos were written following each interview to document salient themes. 

A grounded theory approach [54] was used to create inductive codes for the initial codebook. First, six purposively selected transcripts were coded line-by-line. The purposive sample included four females and two males that held different occupations, had different legal statuses, and were between 33 and 46 years old. The line-by-line coding generated an initial list of codes that were then grouped by themes, including workplace experiences. The first author (Perez-Lua), principal investigator (Young), and a third research team member coded all interviews and met on a weekly basis to discuss the coding process, resolve discrepancies, and refine the codebook as needed. The final codebook included workplace codes that contained narrative descriptions of respondents’ workplace and occupational experiences, including workplace safety conditions, employer policies, employee practices, and personal workplace safety concerns. 

The study team shared the overall findings with the community advisory board and at community report-back meetings, and community organizations provided input on policy recommendations. A more detailed description of the study design, recruitment process, and data collection is provided in the overall study manuscript [50].

### 2.4. Sample Selection

We selected 14 coded interviews from the CLIMA study that were conducted with Latinx immigrants who indicated they were employed as farmworkers at the time of the study. An additional 2 respondents from the CLIMA study had been farmworkers previously but were not included in this sample, as one was retired due to disability and the other had left the industry many years prior to the study.

### 2.5. Analysis

Data for this study was drawn from questions and codes that related to the workplace experience. Diagrams of the relationships between workplace-related codes were constructed to organize workplace codes into groups that could be analyzed together. Code groups were read and analyzed in an iterative fashion. Analytic memos were written to document key themes across interviews. The analytical results from the excerpts were contextualized in the original interviews by returning to the full transcripts and post-interview memos several times throughout the analytical process. Analysis of interview transcripts highlighted organizational aspects and socioeconomic and political conditions that shaped enactment of COVID-19 preventive practices at work. We applied political economy of health theory to connect the structural elements that emerged from farmworkers’ descriptions of their workplace and occupational experiences during the pandemic and contextualize the workplace experiences of respondents in the history and contemporary organization of the agricultural industry. Analytical results also highlighted the impact of the workplace safety climate on workplace camaraderie. Each respondent’s legal status and occupation is presented with their pseudonym to situate their experience in a political context. As demonstrated by the political economy of health theory, the relationship between the agricultural industry and broader social and political conditions, particularly immigration policy, has implications for the occupational safety and overall health of Latinx farmworkers.

## 3. Results

Table 1 summarizes the sociodemographic characteristics of the fourteen respondents in this study. Eleven respondents were employed on crop farms (e.g., vineyards and berry farms). Two respondents were dairy workers: a milker and a veterinary assistant. One respondent was a crop sorter in a fruit-packing warehouse. Two respondents were mayordomos. Mayordomos are crew supervisors who are tasked with recruiting workers, supervision, and overseeing job tasks. Half of all respondents reported that their legal status was undocumented (*n* = 7); two were lawful permanent residents (LPR). 

### 3.1. Overview of the Themes

Regulatory oversight and existing occupational health and safety mechanisms strengthened COVID-19 workplace safety by enforcing social distancing, mask use, and other safety behaviors (Theme 1). In the absence of regulatory oversight and enforcement, COVID-19 workplace safety was largely influenced by the economic interests of employers (Theme 2). Employers often delegated workplace safety to mayordomos (i.e., crew supervisors). Mayordomos were also expected to meet the economic demands of their employer (Theme 3). Caught between their employer’s economic interests and farmworkers’ workplace safety concerns, mayordomos were often empowered by their managerial position to use intimidation tactics and coerce workers into laboring without protections. These organizational conditions normalized the worker—rather than employer—responsibility for workplace safety. Farmworkers exercised individual agency to enact personal COVID-19 safety measures within the organization of the agricultural industry (Theme 4). However, financial hardships were a barrier to purchasing personal protective equipment (PPE) or taking time off from work to quarantine; respondents who reported they were undocumented could not access pandemic-related assistance. Respondents who could or wanted to enact personal safety practices were sometimes limited by workplace arrangements and job tasks that required proximity to other workers or that made it difficult to wear a mask (Theme 5). As a result, there was an unequal implementation of COVID-19 safety measures among farmworkers in the workplace (Theme 6). Farmworkers became suspicious and afraid of workplace COVID-19 infections. Divisions formed between workers who could enact safety practices and those who were incapable of or unwilling to, negatively impacting workplace comradery. 

The next sections provide thick descriptions of how these organizational aspects and socioeconomic and political contexts shaped farmworkers’ COVID-19 workplace safety.

### 3.2. Theme 1: Regulatory Oversight and Access to Protections

Respondents’ workplace safety descriptions indicated that regulatory oversight served as a critical structural intervention to protect their occupational health and safety during the COVID-19 pandemic. Some respondents reported that state-level mask policies, onsite safety inspections, and occupation-specific safety standards expanded their access to COVID-19 protections. State policies that mandated employers to provide PPE granted respondents with access to masks and sick pay. For example, the Executive Order N-51-20, or “COVID-19 Supplemental Paid Leave” policy, required agricultural employers on large farms to provide COVID-19 sick pay to workers [55]. Juan (crop farm, LPR), a mayordomo, described how the policy allowed him to reassure sick workers with financial concerns. He said,

“One time, a young worker arrived sick, and he got angry at me, but I told him to go home. I told him, ‘Look, just go home.’ [The worker said,] ‘But who is going to pay this, pay that?’ [I said,] ‘If you have COVID, they will help you. They will pay you your 80 h.”

Social distancing and mask policies were enforced when workplaces were subject to onsite safety inspections during the pandemic. Victoria (crop farm, undocumented) explained that her employer’s insurance company enforced social distancing between workers during worksite safety inspections. Juan (crop farm, LPR) said he wore his mask when he saw “the [safety inspector] car arrive” at his worksite to avoid a workplace safety violation citation. Juan communicated the difficulty of wearing a mask in the extreme heat to the field inspector and he received guidance for how to manage the two safety hazards. 

Workplaces that received routine safety training during the pandemic also received COVID-19 safety information in addition to other occupational health information. Juan (crop farm, LPR) described how the California Occupational Safety and Health Administration (Cal/OSHA) expanded its training focus due to the pandemic:

“Every year we attend an 8-h class, and they talk to us about sexual assault in the workplace, hot weather—all the implications of the job. Before, Cal/OSHA visited our workplace to confirm that we had clean bathrooms, that the workers had what they needed, like their hat; to check that you treated the workers well, if you are paying workers…but this year they focused on COVID.”

Occupational safety standards and practices that were in place prior to the pandemic facilitated adoption of new COVID-19 safety practices. For instance, Castro (dairy, undocumented) was a veterinarian assistant and regularly used PPE because he faced biological hazards in his occupation. During the pandemic, he began to use masks for protection against COVID-19 in addition to his regular PPE. In contrast, Rogelio (dairy, undocumented) was a milker who had never used PPE in his occupation, nor did he receive masks or gloves during the pandemic. 

In sum, some agricultural workplaces were subject to occupational health and safety regulatory oversite, statewide public health orders, and existing mechanisms for workplace safety. These policies and processes facilitated the implementation and enforcement of COVID-19 workplace safety practices during the pandemic; in some cases, they protected farmworkers’ access to COVID-19 protections that fostered safe working conditions. However, as we discuss below, organizational aspects of the agricultural industry produced unsafe workplace conditions that went unchecked due to unequal regulatory oversight across worksites and occupations.

### 3.3. Theme 2: Employer Economic Interests and Decisions about Workplace Safety

Although regulatory oversight and health policies promoted safe working conditions in some workplaces, organizational aspects emerged as primary factors shaping COVID-19 workplace risks and safety. Most respondents reported that it was primarily employers who made decisions about workplace COVID-19 safety. Those decisions were driven by employers’ economic interests. Juan (crop farm, LPR) explained,

“Here in the [agriculture] fields we don’t have the same rules… The more the [growers] save, the more they earn…they don’t want to spend [on safety].”

As a result, respondents’ access to COVID-19 protections was inequitable across worksites. While some respondents reported that their employer provided masks and sick pay, other respondents only received hand sanitizers or educational materials. Some respondents said they did not receive any COVID-19 protections or information from their employer. Maria (crop farm, LPR) remarked, “in the fields they don’t provide masks”. 

Respondents reported feeling that employers viewed farmworkers as disposable—a “farm tool and nothing more”, as Rogelio (dairy, undocumented) put it. Rogelio said employers preferred to replace sick workers rather than invest in farmworker health. He elaborated,

“[Employers] are not interested in protecting our health…We are workers and that’s it. [Employers] must think that there is no shortage of those who will come asking for work if one or another gets sick.”

In the absence of regulatory oversight and enforcement, the organization of the agricultural industry empowered employers to make workplace safety decisions that were influenced by their economic interests. Respondents reported that employers’ workplace safety decisions determined their access to PPE and sick pay, resulting in inequitable access to COVID-19 protections among farmworkers. The absence of employer-provided COVID-19 protections left respondents with a sense of disposability in the eyes of their employers. 

### 3.4. Theme 3: Mayordomos and Workplace Safety 

Employers delegated workplace safety to mayordomos (agricultural supervisors and foremen). Consequently, mayordomos played a central role in shaping workplace safety during the pandemic. Their managerial position required them to enforce both workplace safety policies and their employers’ economic demands. Two mayordomo respondents described that safety policies and worker protections were incompatible with productivity standards. As they explained, high productivity standards often created workplace conditions that farmworkers perceived to be unsafe or unfair. Mayordomos reported feeling caught between their employer’s demands and farmworkers’ concerns about their working conditions. Juan (crop farm, LPR) explained this difficult position in which he found himself. He said,

“[When] the boss sees green fruit, the mayordomo is the first person he scolds…Sometimes he asks you to suspend that worker [who picked green fruit], or to not bring him back the next day. It is difficult sometimes….” 

Respondents who labored under a mayordomo further highlighted the tensions that resulted when mayordomos had to enforce productivity standards during the pandemic. In some workplaces, mayordomos used intimidation tactics to enforce productivity and minimize farmworkers’ COVID-19 safety concerns or shut down their PPE requests. Maria C (crop farm, LPR) stated, 

“The mayordoma just tells us to use masks, to use the face coverings, but we have told her, ‘The boss has to give us masks, or make sure we’re not crowded.’ But she says, ‘Well, if you have the sufficient means, I advise you to buy your own farm so that you can be your own boss and do whatever you want, because here we come to work and obey.’ And that’s all she told us, basically, that we don’t have a choice.”

Respondents also described how mayordomos’ organizational position empowered them to use intimidation tactics to enforce employers’ economic interests. For example, Jovie (crop farm, unknown) and Juana (fruit sorter, undocumented) shared that their mayordomos attacked their work ethic and threatened to withhold their wages or end their employment when they voiced concerns about workplace safety. Jovie recalled, “The mayordomo [told us], ‘The person who wants to work can get their tools and start working. And whoever doesn’t want to, well, simply leave.’” 

Some mayordomos enforced COVID-19 workplace safety practices, such as mask use and sanitation, without resources or the support of their employer. Luis (crop farm, naturalized) shared an example of the tensions that arose between mayordomos and farmworkers who were frustrated with the enforcement of COVID-19 safety policies without the provision of PPE:

“And I tell [the mayordomo], ‘All you do is tell us [to take COVID-19 precautions] but you don’t give us the tools we need. Since we started working, you told us there would be disinfectant in the bathrooms. The bathrooms don’t even have water to wash our hands, they don’t have toilet paper, they don’t have anything’. And then [the mayordomo] would just stand there, quietly, and didn’t know what to say.”

One mayordomo avoided tensions with his crew by sharing his personal experience with COVID-19 to encourage mask use in the workplace without enforcing it as a policy. He distinguished himself from other mayordomos who used intimidation tactics to enforce COVID-19 safety policies. Eli (crop, undocumented) recalled,

“The mayordomo called a meeting with all the workers and he pleaded with them to wear their masks, because one of his workers died [of COVID-19]. He said, ‘Please wear your masks. I am asking you as a favor, not to be impolite, and I don’t want to be rude, because you know what mayordomos in other crews are doing. They are stopping those without a mask and sending them home.’” 

As respondents’ experiences illustrate, mayordomos’ organizational position and role in advancing their employers’ economic interests shaped the workplace safety. Their approach to COVID-19 workplace safety policy was influenced by their employers’ investment in workplace safety. As a result, the organization of the agricultural industry often normalized unsafe workplace conditions and normalized the worker—rather than employer—responsibility for workplace safety.

### 3.5. Theme 4: Farmworker Agency and COVID-19 Practices in the Workplace

Within the organization of the agricultural industry, respondents exercised individual agency to enact personal COVID-19 safety practices at work, such as wearing a mask and social distancing. However, financial barriers rooted in broader socioeconomic and political conditions further limited respondents’ individual agency to enact personal COVID-19 safety practices at work. The precarious nature of agricultural employment contributed to farmworkers’ financial insecurity. The agricultural industry faced supply chain delays, higher costs of production, and economic losses as it adapted to organizational shifts in safety policies [56]. Several respondents said that pandemic disruptions to agricultural production and exportation resulted in their sudden loss of work. Legal status exclusions from critical pandemic aid and other economic relief [57] further exacerbated respondents’ financial insecurities. 

Severe financial conditions left many respondents with no choice but to continue working during the pandemic despite the unsafe working conditions and limited access to COVID-19 protections. Maria C (crop farm, LPR) explained,

“If I go home, I take care of my health. But then, who is going to feed me at home? We either die of hunger, or we die of the virus.”

Because of financial pressures, travel to work sites was fraught with risk. Some respondents had few transportation options and had to carpool to work sites, which increased their close contact with others and risk of exposure to a COVID-19 infection. Jesus (crop farm, unknown) shared,

“And right now, I am carpooling with my brother, because we only have one car. I leave the car at the house because sometimes my wife needs it to run household errands, and that is why I don’t drive to work, I carpool with my brother right now.”

Victor (crop farm, unknown) was a mayordomo and explained that workers who did not have their own vehicle carpooled with others. Being in an enclosed space with multiple people made some respondents feel at risk of exposure to a COVID-19 infection. Juana (fruit packing, undocumented) suspected that many people in her workplace had been infected with the novel coronavirus while carpooling to work. 

Financial hardships continued to reduce farmworkers’ individual agency to enact personal COVID-19 practices once they arrived at work sites. Maria C (crop farm, LPR) explained that financial barriers to purchasing PPE contributed to the low adherence to the mask mandate in her workplace. Juan (crop farm, LPR) said some workers were unwilling to stop working even if they had COVID-19 symptoms because they worried about the potential loss of income. He shared,

“There are workers who don’t receive anything, like those without papers, and they worry about, ‘How am I going to work if I’m sick? How am I going to pay for the bills? How will I buy food for my kids?’”

Indeed, the precarious nature of employment and legal status exclusions from economic aid contributed to farmworkers’ severe financial conditions. Farmworkers faced economic barriers to enacting personal COVID-19 practices in the context of organizational aspects that normalized worker responsibility for workplace safety.

### 3.6. Theme 5: Job Tasks, Work Arrangements, and Workers’ Risk Perception 

Even if respondents could access COVID-19 protections, either through their employer or personal means, their individual agency to enact personal COVID-19 safety measures was often influenced by their job tasks and workplace arrangements. Job tasks and work arrangements were not always designed to accommodate variable occupational health and safety risks.

Specific job tasks and workplace arrangements posed unique challenges to workplace COVID-19 safety. Rogelio’s (dairy, undocumented) occupation as a milker required him to ride in a tractor with another worker for long periods of time. Although he wore his mask, Rogelio felt that sharing an enclosed space with his coworker increased his risk of infection. In contrast, the veterinary assistant, Castro (dairy, undocumented), shared an outdoor workspace with three other workers. They all worked individually for most of the workday and could practice social distancing. On crop farms, the spacing between crops supported social distancing but weather conditions made it difficult to wear a mask. Luis (crop farm, naturalized) shared that the separation between trees on the farm kept workers far apart, but the intense summer heat made it difficult to wear a mask while he worked. 

Some farmworkers perceived their job duties and work arrangements put them at risk of infection regardless of their access to protections or ability to enact personal COVID-19 safety practices. For instance, Juana (fruit packing, undocumented) understood that crowds increased the risk of exposure to COVID-19. However, she could not avoid crowding inside the fruit packing warehouse. She described how hundreds of workers sorted, packed, and transported fruit inside the warehouse. Although her employer installed plastic barriers between sorters, spaced each sorter “more-or-less” six feet apart, and dismissed workers for lunch “two lines at a time” to reduce crowding in the kitchen, Juana still took the initiative to socially distance outdoors during her lunch break due to fear of exposure to COVID-19. She shared,

“What we would do, instead, was eat our lunch outside, where we could be by ourselves. Out of fear, you know. You see that crowds bring a lot of risk, of getting infected, so instead we would go outside and eat under the little trees.” 

Luis (crop farm, naturalized) reported feeling at increased risk of exposure to infection at the end of the day when farmworkers had to come together to complete the final crop row. Luis made an effort to avoid his coworkers who did not enact personal COVID-19 safety practices, but it became more difficult during this part of the workday. 

As these experiences show, respondents’ individual agency to enact personal COVID-19 practices was often determined by whether workplace arrangements and job tasks could accommodate COVID-19 safety measures. Workplace arrangements shaped respondents’ perceived risk of exposure to a COVID-19 infection at work.

### 3.7. Theme 6: Personal Responsibility vs. Workplace Camaraderie

Not only did organizational aspects shape respondents’ safety from COVID-19 in the workplace, but they also impacted their relationships with other workers. The lack of organizational accountability to maintain safe working conditions resulted in the unequal implementation of workplace safety behaviors among workers due to differences in personal access to protections and individual agency to enact safety measures. As a result, farmworkers became fearful and worried about COVID-19 spreading in the workplace when they perceived their workplace conditions put them at risk of infection. Victoria (undocumented crop farmworker) commented,

“I will say that COVID did impact me. Because one doesn’t even trust your coworkers and they don’t trust you. We are all here, afraid.”

Fear and distrust drove divisions between workers who enacted personal COVID-19 safety practices and those who did not. Workers who wore masks and socially distanced formed their own working groups and excluded workers who they observed did not practice COVID-19 safety. Luis (crop farm, naturalized) explained,

“When we were almost done with the day, we teamed up with three or four others to help [finish the crop row]; but we were just with those in our crew, because we knew that we took care of ourselves…” 

Some farmworkers viewed those who did not enact safety practices as irresponsible or malicious. They socially excluded coworkers who were suspected of having COVID-19. For example, Eli (crop farm, undocumented) felt ostracized by her coworkers when she showed signs of a COVID-19 infection but was unable to obtain a COVID-19 test and was asked to return to work. She shared,

“And people didn’t even want to talk to me because they said, ‘This one is probably sick. She probably has COVID.’ I even heard a coworker say, ‘If a worker gets sick and comes to work, that’s now a crime…Because they know that they’re sick and can infect others, and we have the right to call the police so that they can come arrest them and take them to get checked by a doctor because that’s now a crime.’ And I froze and thought, ‘Why do they think that?’ It was something complicated and also sad.”

A sense of distrust among farmworkers made it difficult for some to communicate their safety concerns or needs to their coworkers. For example, Victoria (crop farm, undocumented) lied about why she was setting physical boundaries at work. She explained, 

“They ask me, ‘Are you angry? Bitter?’ And I tell them, yes, because I feel embarrassed to tell them [that I want to social distance]…But really it is because I do not let people near me. Because I have children at home, and if I do not take care of myself, who is going to take care of my kids?”

Under the unsafe workplace conditions that were normalized by the organization of the agricultural industry, a sense of distrust and the fear of COVID-19 in the workplace divided workers. These conditions negatively impacted workers’ ability to communicate their safety concerns and needs to each other and they dissolved workplace comradery. 

## 4. Discussion

Through the lens of political economy of health theory, this qualitative study of Latinx farmworkers’ workplace safety during the COVID-19 pandemic sought to identify and describe how organizational aspects of the agricultural industry and broader socioeconomic and political conditions shaped COVID-19 workplace safety. A political economy of health approach to farmworker health contributes insight into the structural mechanisms that produce health risks [58]. Our study demonstrated how the organizational aspects of the agricultural industry’s intersection with broader socioeconomic and political conditions produced COVID-19 workplace safety risks for farmworkers by limiting access to COVID-19 protections and reducing farmworkers’ individual agency to enact personal safety measures against COVID-19. These conditions promoted presenteeism (i.e., working while sick) and workplace environments that were conducive to viral transmission (i.e., working without masks) [59]. Additionally, organizational conditions negatively impacted workplace comradery. Workplace comradery is essential to collective agency and achieving safe working conditions through collective bargaining [60]. These findings add depth to epidemiological research on COVID-19 among farmworkers that have shown clear disparities in farmworker health by highlighting the organizational mechanisms, processes, and dynamics that may have increased farmworkers’ risk of exposure to a workplace COVID-19 infection [3,4,39].

As demonstrated by the findings of this study, state and federal labor policies—when enforced—can protect workers’ safety from COVID-19 at work. There is a need to expand regulatory enforcement of occupational safety standards in the agricultural industry to ensure that all agricultural employers are held accountable to workplace and occupational safety policies and standards. Other studies have similarly shown that limited safety enforcement and inadequate standards have negative health consequences because safety policies go unenforced in the absence of regulatory intervention [61].

In the absence of regulatory oversight, the organization of the agricultural industry—which reflects a centuries-long history of growers benefiting from racist policies and capitalist structures to advance their economic interests—functioned to prioritize profit at the expense of farmworker health and safety during the pandemic. Respondents perceived that the organizational structure of the agricultural industry empowered employers to make workplace safety decisions that were driven by their economic interests. These findings are consistent with previous studies that found employers’ decisions about workplace safety shaped farmworkers’ access to workplace safety information, safety training, and personal protective equipment [62,63,64]. Previous studies have also shown that farmworkers perceive that their employers prioritize economic interests over worker health and safety [34]. Regulatory oversight and enforcement are needed to prevent employers from solely shaping workplace safety conditions based on their economic priorities, as it can produce workplace conditions that are conducive to viral transmission. 

Employers often delegated workplace safety to mayordomos, who found themselves caught between their employers’ economic interests and farmworkers’ workplace safety concerns. Some mayordomos (crew supervisors) were empowered by their managerial position to coerce workers to labor without COVID-19 protections. Other mayordomos faced challenges implementing COVID-19 safety policies without resources. Mayordomos who were interested in protecting farmworkers’ health and improving working conditions were disempowered by the lack of access to resources needed to foster a working environment that could support the enactment of COVID-19 preventative practices, contributing to the tensions that arose between mayordomos and workers. Consistent with political economy of health theory, these findings show that organizational roles that exist to serve institutional capitalistic aims are incompatible with occupational health [12]. Studies in the past have emphasized the impacts of harmful managerial practices, supervisory control, and workplace hierarchies on the health of farmworkers [62,63,65]. Several studies suggest that providing supervisors with knowledge about safety, labor policies, and worker rights can improve workplace safety for farmworkers [66]. However, the findings of this study show that the organizational role of the mayordomo primarily serves to implement employer priorities regardless of the impact on worker health. This finding provides insight into why promotoras—lay health promotors—and outreach workers face limitations to fostering safe working environments even though their roles are designed to support and empower farmworkers [67]. The introduction of roles that aim to improve farmworker health cannot succeed if the overall organizational structure serves the economic interests of employers. 

The precarious nature of agricultural employment contributed to respondents’ severe financial conditions. Legal status exclusions from critical federal and state assistance [57] exacerbated farmworkers’ financial hardships. Financial pressures deprived respondents of their right to choose to work and limited respondents’ personal access to COVID-19 protections, ultimately diminishing their bodily autonomy within the organization of the workplace. Financial pressures and economic barriers to personal COVID-19 protections, such as sick time, promote presenteeism [59]. Presenteeism is particularly important to address among farmworkers, as the majority are excluded from employment benefits such as unemployment benefits and disability because of their undocumented legal status. The organization of the agricultural industry (e.g., regulatory oversight, employers, and supervisors) must promote the use of sick pay, as undocumented farmworkers are largely excluded from the labor market and safety net benefits, contributing to their job loss and financial concerns that make them hesitant to take sick time off even when it is available.

Our study revealed that organizational-level inequities in occupational health and safety enforcement reproduced inequities in workplace safety among workers, as the onus to ensure workplace safety fell on individuals who did not have equal access to COVID-19 protections. These inequities, in turn, impacted workplace comradery as workers distrusted each other and feared a workplace infection. The impacts on workplace comradery may lead to reluctance to collaborate and collectively advocate for workplace safety protections from employers. Although farmworkers continue to be excluded from the National Relations Labor Act, which allows workers to form unions, collective power can still be an effective tool to achieve workplace protections [68]. In recent years, California has expanded protections for farmworkers through the enactment of the Agricultural Labor Relations Act (ALRA) of 1975 and Assembly Bill 2183 (AB2183) of 2022 [69,70]. The ARLA grants farmworkers the right to collective bargaining. AB2183, which resulted from a farmworker-led social movement that pressured the state to expand labor protections to farmworkers, allows farmworkers to vote for a union by mail-in ballots to avoid employer retaliation. Despite California’s expanded protections for farmworkers, our analysis reveals that the organization of the agricultural industry maintains workplace safety inequities that potentially reduce farmworkers’ collective agency to achieve safe working conditions through collective bargaining.

Farmworker health research should give attention to how broader discriminatory policies and sociopolitical climates uphold power inequalities in agricultural workplaces and how they might contribute to farmworker occupational health disparities. This study took place in California, a state that has led the country in strengthening occupational health standards in agriculture and expanding protections to immigrants and farmworkers [71,72,73]. Yet, as this study has shown, farmworkers continue to experience inequitable access to protections and resources that are critical to their occupational and overall well-being. Research on contemporary guestworker programs, such as the U.S.’s H-2A visa program and Canada’s Seasonal Agricultural Worker Program (SWAP), show that these programs uphold power inequalities that serve capitalist agriculture at the expense of migrant, Black, and Latinx farmworkers [31,74]. Other farmworker health studies have underscored the need for immigration policy to provide undocumented farmworkers with a clear pathway to citizenship and empower them with access to worker rights and protections and the social safety net [9,75,76]. However, the history of the agriculture industry suggests that the agricultural industry continuously benefits from discriminatory policies that disenfranchise whole populations, including those with legal status. Future studies should examine how broader policies and social processes shape the vulnerability of the hired agricultural labor force as the sociodemographic composition of the labor force changes. For example, Indigenous workers are a growing subpopulation of the agricultural labor force [77,78]. As Holmes [79] demonstrated, racial and ethnic hierarchies are reproduced within the agricultural labor force, and Indigenous farmworkers are often relegated to the bottom of the workplace hierarchy. Failing to address the reproduction of social hierarchies in the agricultural workplace will continue to perpetuate a cycle of poverty and risks to health among the populations in the agricultural labor force. 

### 4.1. Policy Recommendations

Policies to prevent abusive conduct in the workplace should target employers’ economic interests. Educational interventions should be supported by financial penalties and prosecutory action that hold agricultural employers accountable for noncompliance with safety standards. Fines should have a greater economic impact on growers than the cost of investing in workplace safety. Changing the economic incentives may pivot the mayordomos’ role towards enforcing safety as it becomes a critical aspect in mayordomos’ function to protect the economic interests of the grower. When new hazards are introduced in the workplace, such as COVID-19, fines for workplace safety violations should be increased to support farmworker safety against additional health risks. 

When implementing policies that rely on employers to comply with safety standards, such as COVID-19 safety policies or the Worker Protection Standard (WPS), public health and regulatory safety agencies, such as the Federal Emergency Management Agency (FEMA) and OSHA, should supply agricultural workplaces with the resources and materials to support workplace safety. When occupational health policies cannot enforce employer compliance with occupational health policies, policy makers should create channels of communication that empower workers to directly request PPE for their workplace. This strategy may help mitigate the impacts of employer attitudes towards safety in shaping farmworkers’ access to protections, such as PPE. 

Funding towards OSHA and state-level regulatory agencies must be increased to support staffing and enhance the agency’s enforcement capacity. OSHA enforcement jurisdiction must be extended to all farms and agricultural facilities. To ensure that all farmworkers are protected at work, occupation-specific safety standards should be set and enforced for all agricultural occupations regardless of farm size or operations. Currently, only farms that employ 11 workers or more, or who employ temporary guestworkers, are subject to OSHA inspections and citations for workplace safety violations [80]. The number of hired OSHA inspectors is at a historic low [81]. Setting and enforcing occupation-specific standards equally across industries and workplaces may protect the health of farmworkers who rotate between workplaces and occupations throughout the year as they follow the agricultural seasons. It may also help foster an overall stronger culture of safety within the agricultural industry. 

Lastly, as Furton and colleagues [82] suggest, the COVID-19 pandemic has brought opportunity to make structural changes because institutions become more malleable as they respond to health crises. This study highlighted the organizational aspects of the agricultural industry and workplace that require structural changes to support the health of farmworkers. While our policy recommendations have the potential to improve workplace safety in agriculture, to achieve transformational structural changes, the agricultural industry must shift its focus from profit to feeding communities [45,46]. 

Moreover, addressing workplace safety inequities in agriculture requires policymakers to address broader social inequities rooted in structural racism. Comprehensive policies across policy areas (e.g., healthcare, education, environment) are necessary to address the intersectional impacts of racism, xenophobia, capitalism, and other forms of exclusion that make Latinx immigrant farmworkers vulnerable to industry exploitation [83]. 

### 4.2. Strengths and Limitations

A strength of this study is the diversity of farmworker positions, workplaces, and work structures presented. This study was able to identify organizational aspects that may shape the health of farmworkers across industries while highlighting worksite-level and occupational-level mechanisms by which these organizational factors may shape health. Another strength of this study was that the study sample focused on the experiences of farmworkers who lived and worked in rural regions of California. California is the leading agricultural-producing state [84]. Yet, California farmworkers are underrepresented in farmworker health studies. Most farmworker health studies are either conducted in an eastern region of the U.S. or they do not specify a region [85].

Limitations of this study include its focus on analyzing the experiences of a small sample of farmworkers and the lack of direct perspectives from growers and occupational health agencies. Additionally, the study sample was collected over the span of the first year of the pandemic. The experiences of farmworkers who were interviewed during the early stages of the sampling process may have been different from the experiences of farmworkers who were interviewed during a later stage in the sampling process due to the rapidly changing pandemic conditions. The experiences of the respondents in this study are unique to the Latinx immigrant farmworker community and study regions. The workplace processes and dynamics that were observed in this study may vary from other qualitative studies on farmworker health due to differences in agricultural industries, occupations, geography, and the sociodemographic composition of the sample. Despite limitations, this qualitative study engaged respondents in in-depth conversational interviews that generated rich and detailed descriptions of their experiences, perspectives, and concerns that are likely experienced by other farmworkers across the industry and country who labor under similar socioeconomic, political, and organizational conditions.

## 5. Conclusions

Organizational aspects of the agricultural industry and broader socioeconomic and political conditions influenced farmworkers’ access to COVID-19 protections, such as masks and sick pay, and their individual agency to enact personal COVID-19 safety practices at work. These organizational conditions produced inequities in access to COVID-19 workplace safety protections across worksites and individuals. Under these conditions, farmworkers feared contracting a COVID-19 infection in the workplace and distrusted their coworkers, negatively impacting workplace camaraderie and potentially reducing farmworkers’ ability to achieve safe working conditions through collective bargaining. Policies must address the structural determinants of farmworker health to protect farmworkers from COVID-19 and other existing and emerging occupational health and safety risks.

## Figures and Tables

**Table 1 ijerph-20-07138-t001:** Sociodemographic characteristics of the study sample (N = 14).

Characteristic	N or Mean (SD)
**County**	
Fresno	6
Merced	2
Tulare	6
**Farm**	
Field crop	11
Dairy	2
Fruit-packing warehouse	1
**Gender**	
Female	6
Male	8
**Mean years living in the U.S.**	22 (11)
**Highest level of education**	
High school or less	11
College degree	1
Unknown	2
**Marital status**	
Single	2
Married or living with a partner	10
Unknown	2
**Mean household size**	5 (3)
**Citizenship**	
U.S. citizen	1
Lawful permanent resident (LPR)	2
Undocumented	7
Refused or unknown	4
**Food security ^1^**	
High	1
Marginal	4
Low or very low	7
**Ever used…**	
Social Security	1
Unemployment	3
Disability insurance	0
Worker’s compensation	3
**In the past 2 years, anyone in the household used…**	
Food stamps (CalFresh), Medi-Cal, or Special Supplemental Nutrition Program for Women, Infants, and Children (WIC)	11

^1^ Measured using the USDA’s six-item food security module [53].

## Data Availability

Due to confidentiality and privacy concerns of the participants, the data for this study is not publicly available. Interested individuals may contact the corresponding author to request selected excerpts represented in this study. The excerpts are available in English or Spanish.

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
