# Peer review of "The Influence of Organizational Aspects of the U.S. Agricultural Industry and Socioeconomic and Political Conditions on Farmworkers’ COVID-19 Workplace Safety"

_ijerph, 2023, doi:10.3390/ijerph20237138_

Round 1

Reviewer 1 Report

Comments and Suggestions for Authors

This study seeks ground the unequal experiences and burden of health risks among California farmworkers during the COVID-19 pandemic within a political economy framework. The authors draw from in-depth interviews to highlight several organizational aspects of the agricultural industry that shape workplace safety among farmworkers during the pandemic. The paper represents a contribution to the emergent, yet sparse, literature on the experiences of farmworkers specifically during the COVID-19 pandemic. Strengths include a focus on structural determinants of health – including the history of agricultural labor in the US, organization of work, and use of in-depth interviews to elicit the perspectives of workers themselves. The paper is very well-written. Most of my comments are minor, but there are a few issues that, once addressed, could improve the manuscript further.

Introduction

The literature review/introduction focuses on the political economy of farm worker health. This is a succinct summary of the history, but this section could be improved by further synthesizing how the organizational aspects of the agricultural industry shaped the experiences of farmworkers during the pandemic. Currently, the opening paragraph discusses disparities in COVID-19 risk and rates in agricultural workplaces, but the dots are not connected among the political economy history and early COVID-19 conditions. I would suggest a brief subsection after the political economy section that connects these prior conditions to the exacerbations caused by the pandemic. This section could include some of the emerging literature on the COVID-19 and farmworker health and mental health outcomes. I believe this contextual information about what is known vs. not known would help further emphasize the need for this study and aid with the interpretation of the results.

Additionally, the literature review currently reads as if farmworkers have no agency. While their agency is severely constrained by the political economy outlined in this section, it may be worthwhile to give this discussion a little more nuance by highlighting acts of resistance and organizing despite the structural limitations. This also seems a focus in the abstract that is not fully developed throughout the manuscript.  Indeed, the results seem to describe at least some acts of agency among farmworkers, amidst an array of constraints (section titled farmworker agency and COVID-19 practices in workplace). It seems like there is an opportunity to foreshadow these results as well as some of the information presented in the discussion in the introduction.   

Methods

Line 117: More rationale is needed for the subset of 14 interviews from the larger CLIMA study. How big was the parent study? Were only 14 out of the entire sample farmworkers? It states that those who were not employed in agriculture at the time of the study were excluded, but I am left curious as to if perhaps other participants in the larger study were former farmworkers (and if the pandemic had any effect on their job security etc…). Just a little more information about this rationale would help clarify these questions.

Line 121: Can the authors provide a little more information on their community-engaged approach?

Analysis section: More information is needed to defend how this is a grounded theory approach. While several GT approaches/methods were used to analyze the data, it seems like the study was strongly guided by political economy theory from the beginning and thus the authors fit the data to theory rather than it being generated from the data itself. More clarification and precision in how this converges or diverges with a GT approach would be helpful. There are many variations of GT, so more information about these decisions would strengthen this section.

A researcher positionality statement would help the reader with the interpretation of the results and analysis. This may also help clarify the point above about the theoretical lenses used and how they informed the analysis process.

Results

The paragraph directly after table 1 is slightly confusing. It seems that the text before the table is describing the content in the table, but then the text directly after the table is describing overall themes/findings, without any transition. I recommend the authors revisit the organization here and consider if a subheading after the table where the qual results are briefly summarized would help improve the readability.

Throughout, the authors present in parentheses the workplace and citizenship status of each participant. Yet, the rationale for presenting this is unclear. As a reader, I can imagine why these demographic characteristics are important, but it would be helpful to have the author’s rationale/justification and how these characteristics may shape experiences/vulnerabilities.

The citations within the results section (lines 199, 202) seem out of place. Suggest moving to discussion.

Some of the findings seem to be contradicting each other, which from my perspective is fine as long as they are called out in a way to make sense of discrepant findings. For example, in the discussion (line 455) the authors say that respondents perceived a lack of regulatory oversight empowered employers to advance their own economic interests over that of the workers. However, there is a whole section in the findings (section 3.4 regulatory oversight and access to protections), that seem to contradict these statements. Both can be true, but more nuance is needed in how these are presented in the results and in the subsequent discussion. I suggest re-ordering the way the themes are presented to put these two themes in conversation, e.g., 3.2 “employer economic interests and decisions about workplace safety” and “regulatory oversight and access to protections.” This seems like an opportunity to argue one of the main points of the manuscript about how even though there is some regulatory oversight at the state level (especially in California), the organization of farm work makes it so that these are hard to enforce and easy for employers to get around.

Lines 472-474: While this is still federally the case, is it true for California and the Agricultural Labor Relations Act and AB2183?

The discussion is fitting and overall does a nice job a bringing the literature in to compare with the findings as well as emerging areas for farmworker research (e.g., indigenous populations). You may want to drop the “Latinx” from “indigenous Latinx workers” unless there is a justification for this categorization.

The policy recommendations are a good addition and drive home the point that changing the organization of work is critical for improving farmworker justice – and that piecemeal or individualized approaches that do not consider this will not accomplish their goal. While more practical, feasible, and easy to imagine as possibilities, I do find it interesting that the policy recommendations all fall within and preserve the same capitalist system rather than calling for more radical systemic change.

Reviewer 2 Report

Comments and Suggestions for Authors

The study, based on a qualitative empirical case study methodology, analyzes how organizational and socioeconomic aspects influence the health security of Latino immigrant workers in the agricultural and livestock industry in the U.S. in the context of COVID-19. The topic is emerging, fits with the journal's spirit, and responds to the geopolitical call that seeks to make vulnerable social sectors of the population visible. Nevertheless, I will make some comments to contribute to the discussion and improvement of the manuscript.

1. Title. The title should first prioritize the variables of analysis, then specify aspects of the unit of analysis and contexts. In your work, your variables are three: organizational aspects, socioeconomic aspects, and occupational health. The title even leaves out the variable "socioeconomic" and is also incongruent with your keywords since, in them, you give weight to "health" rather than "occupational safety." Its title should be, for example: "Organizational and socioeconomic aspects that influence the health (or health safety) of Latino immigrant workers in the U.S. agricultural and livestock industry in the context of COVID-19".

2. Structure abstract. The abstract could be better structured along primary lines: brief justification of the problem, explicit objective with an infinitive verb at the beginning, methodology, and findings. Especially note that the aim could be more precise in the current version. Following the observation of point 1, prioritize your objective variables, then a unit of analysis. It could be, for example, "To examine how organizational and socioeconomic aspects influence the health (or health security) of Latino immigrant workers in the U.S. agricultural and livestock industry in the context of COVID-19."

3. Keywords. In your keywords, you give a lot of weight to the "health" variable. It seems to me that, according to the development of his work, rather than "occupational safety," he focuses on the variable of "health safety" or "occupational health."

4. I find the historical framework unnecessary (line 56 and following); however, from a critical dialectic perspective, it could be justified. Explain why the historical context is relevant in this empirical study.

5. Methodology and technique. It would help if you further strengthened the epistemological underpinning of your methodology (I recommend reading Wacker, 1998. Doi: 10.1016/S0272-6963(98)00019-9) and the goodness of its technique with articles specific to its subject, but also with seminal sources such as "Yin, R. K. (1998). The abridged version of case study research. Handbook of applied social research methods, 2, 229-259" and "Lincoln, Y. S., & Guba, E. G. (1990). Judging the quality of case study reports. Internation Journal of Qualitative Studies in Education, 3(1), 53-59", for example. Argue with support from the literature why "qualitative methods are appropriate in studying this phenomenon"; why "qualitative methods provide a critical method"; what other empirical case studies have been based on political and economic theory?

6. Annex to the semi-structured instrument. One of the meanings of the description of an excellent methodological design is that it can be replicable to contrast results in future research. Therefore, it would be very enlightening to include the semi-structured interview as an annex. 

7. In your conclusion, remember to answer the implicit question of your research: HOW do organizational and socioeconomic aspects influence the health (or health security) of Latino immigrant workers in the U.S. agricultural and livestock industry in the context of COVID-19? They only answer about the "what," but there is a need to go deeper into the "how." 

After these changes, it could be considered publishable. Congratulations to the authors for this socially relevant work.

Fraternal,

Reviewer.

Round 2

Reviewer 2 Report

Comments and Suggestions for Authors

The manuscript "The Influence of Organizational Aspects of the US Agricultural Industry and Socioeconomic and Political Conditions on Farmworkers' COVID-19 Workplace Safety," after the suggested corrections, is publishable and relevant to the academic contribution within the spirit of the journal. 

Thanks to the authors for the gift of their research.

Fraternal, 

The Reviewer